# Quality of clinical assessment and management of sick children by Health Extension Workers in four regions of Ethiopia: A cross-sectional survey

Dawit Wolde Daka[1]*, Muluemebet Abera Wordofa[2], Mirkuzie Woldie[1,3], Lars Åke Persson[4,5], Della Berhanu[4,5]

1 Faculty of Public Health, Department of Health Policy and Management, Jimma University, Jimma, Ethiopia, 2 Faculty of Public Health, Population and Family Health Department, Jimma University, Jimma, Ethiopia, 3 Fenot Project, Harvard T.H. Chan School of Public Health, Addis Ababa, Ethiopia, 4 Ethiopian Public Health Institute, Addis Ababa, Ethiopia, 5 The London School of Hygiene & Tropical Medicine, London, United Kingdom

* dave86520@gmail.com

**Data Availability Statement:** Data from this study are co-owned by the participating institutions and stored in a depository at the Ethiopian Public

## Abstract

### Background

Care-seeking for sick children at the Ethiopian primary health care level is low. This problem may partly be due to unfavorable community perceptions of the quality of care provided. There is, however, limited knowledge on the quality of the clinical assessment and management provided by the health extension workers at the health posts. This study aimed to examine the quality of clinical assessment, classification and management provided to sick under-five children by health extension workers in four regions of Ethiopia.

### Methods

Clinical observations of 620 consultations of sick children by health extension workers were conducted from December 2016 to February 2017. A clinical pathway analysis was performed to analyze whether sick children were appropriately assessed, classified and managed according to the integrated Community Case Management guidelines.

### Results

Most sick children presented with complaints of cough (58%), diarrhea (36%), and fever (26%).Three quarters of children with respiratory complaints had their respiratory rate counted (74%, 95% CI 69–78), while a third (33%, 95% CI 27–40) of children with diarrhea were assessed for dehydration. Half (53%, 95% CI 49–57) of the sick children were assessed for general danger signs, while a majority (89%, 95% CI 86–92) had their arm circumference measured for malnutrition. Half of the sick children received some treatment and less than one-fifth were referred according to the integrated Community Case management guidelines. Comprehensive counseling was provided to 38% (95% CI 35–42) of the caregivers.

Health Institute (EPHI). The use of these data is guided by a data sharing agreement that states that data will be made available upon reasonable request but are not publicly available during the period when Ph.D. students and other involved researchers are analyzing and reporting based on these data. Data can be accessed from the secretary of 'Data sharing committee of EPHI-LSHTM Collaborative Projects'. Contact information: Name: Mrs. Martha Zeweldemariam E-mail: martha.zeweldemariam@lshtm.ac.uk

**Funding:** The study was supported by a grant from Bill and Melinda Gates Foundation to The London School of Hygiene & Tropical Medicine (grant OPP1132551). The funder had no role in data collection, analysis, or interpretation of results.

**Competing interests:** The authors have declared that no competing interests exist.

## Conclusion

The Ethiopian health extension workers' clinical assessment, classification and management of sick children did to a large extent not follow the clinical guidelines. This lack of adherence could lead to misdiagnoses and lack of potentially life-saving treatments.

## Introduction

Despite worldwide progress in child health, survival remain an urgent concern. The under-five mortality rate in most low- and middle-income countries remain high [1–3]. As of 2018, on average 15,000 children under the age of five died worldwide every day, mostly from preventable causes. Most of these deaths occur in the poorest nations of the world, where access to health care is limited [3].

Ethiopia made notable progress in reducing child mortality during the past few decades and reached the fourth Millennium Development Goal on child survival [4]. The 2019 under-five mortality rate of 55/1000 live births is, however, still high [5]. For further reduction of child mortality increased utilization of primary child care as well as improved quality of services are needed [4, 6]. In Ethiopia, primary health services for sick children are provided within the integrated Community Case Management (iCCM) program that was initiated in 2010 [7] and the community based newborn care program established in 2014 [8]. These interventions are delivered as part of the Ethiopian primary care services, i.e., the health extension program [9]. In this program, around 37,000 health extension workers have been trained and deployed to almost 18,000 health posts to provide treatment for sick children closer to their homes [10].

Following the Sustainable Development Goals (SDGs), countries have pledged to reach targets on child survival. These goals also include universal health coverage with access to safe, effective, high-quality and affordable care for women, children and adolescents [11]. Child health interventions can be delivered at health facility and community levels. Particularly in resource limited settings, community-based services need to be strengthened so that child health interventions can be delivered by appropriately trained and supported community health workers, who extend the health services to the local community [12–16].

Various aspects of quality of the community-based child health services in Ethiopia have been investigated. A few studies in the early implementation period of iCCM indicated the need for strong leadership and support to the program, as well as the need for improved management of the sick child by the health extension worker. Interventions have been implemented to improve the quality of community case management programs, showing that performance reviews and clinical mentoring were successful [13], while supportive supervision had less evident effect on quality of services [10, 13]. A recent study, based on the same larger survey as this article, showed that the health extension workers' diagnostic accuracy was low [17]. The utilization of child curative services at the health post has also remained on a relatively low level [7]. The reasons for low care-seeking for sick children includes low community awareness about services provided at the health post [6], unfavorable community perceptions about the quality of care provided by the health extension workers [18], inappropriate perceptions of illness severity and low expectations of available treatments [19]. To the best of our knowledge, no previous study has analyzed the entire process of managing children with common illnesses 2–59 months at Ethiopian health posts. Sick children need to be properly

assessed, their illness correctly classified and provided with appropriate treatment and advice in accordance with the relevant guidelines.

Hence, the aim of this study was to assess the quality of the health care provided to sick children mobilized to visit health posts. Specifically, we examined the assessment, classification, treatment, counseling and referral done by the health extension workers at health posts in four regions of Ethiopia.

## Subjects and methods

### Study setting

This study was part of a large baseline survey, which included household and health facility modules and aimed to determine the effectiveness of the Optimizing Health Extension Program (OHEP) intervention in Ethiopia. OHEP aimed to increase utilization of primary health care services for sick children through interventions that engaged communities, strengthened the health system at the primary care level and created ownership and accountability for child health services at the district level. The trial registration number is ISRCTN12040912. This paper was based on a study conducted in selected health posts of 52 districts across four Ethiopian regions: Oromia, Amhara, Southern Nations, Nationalities, and Peoples (SNNP) and Tigray. The selection of the districts was carried out by the regional health bureaus and the researchers had no role. The protocol for the evaluation of the OHEP intervention has been published [20].

The health post is the lowest health care delivery unit in Ethiopia designed to serve a population of 5000 and managed by at least two health extension workers. The health extension worker is a female paramedic professional, who has completed grade 10 and has received an additional one-year training on the 17 packages of the health extension program (S1 Box). The iCCM program advises the health extension worker to examine and manage the child in the following ways: assess for general danger signs; ask for symptoms; assess the child; classify the disease; treat, refer if needed; and counsel the caregiver. The symptoms that should be assessed by this approach are cough, diarrhea, fever, and ear problems. The health extension workers also assess the presence of malnutrition, anemia, HIV and check children's vaccination and vitamin A supplementation status in the health post records and by asking the caregivers. The health extension workers usually manage sick children at the health posts. They bring some medicines and supplies when doing home visits, but the major emphasis when being in the community is on preventive and promotive activities.

### Study design and participants

A facility-based cross-sectional study was carried out involving observation of clinical consultations and interviews. The study participants were children 2 to 59 months of age, who sought care from the health extension workers at the health posts following a mobilization of sick children. The community mobilization was undertaken by the data collection team in collaboration with the local administration.

Firstly, target communities were selected from the 52 study districts using a list of enumeration areas from the 2007 Ethiopian Housing and Population Census as the sampling frame. The cumulative population size across the study areas was calculated and 200 enumeration areas were selected with probability proportional to the population size. Each enumeration area formed a cluster and each cluster constituted a primary sampling unit.

Then, health posts serving the households in the randomly selected clusters were included in the study.

The sample size calculations were done for the forthcoming evaluation of OHEP intervention. Consultations of 800 sick children was planned to be included based on the forthcoming analysis before and after the OHEP intervention. We aimed to include 4 sick children 2–59 months of age consultations with health extension workers per health post to reach a total sample size of 800 consultations. This number of consultations was considered to have 80% power to detect a difference of at least 15 percentage points when comparing the change in the ability of health extension workers to correctly assess, classify and treat diseases at baseline vs. endline in intervention and comparison areas. For the present study, the planned sample size implied that we had a post-hoc statistical power to ascertain a prevalence (for example when assessing general danger signs) of 50% with an uncertainty of 5 percent units, with an alfa of 0.05 and a statistical power of 81%. Given the present study's final sample of 620 sick children consultations by health extension workers, the corresponding figure is 50% with uncertainty of 6 percent units, alfa 0.05 and statistical power of 85%.

## Measurement and data collection procedure

A structured observation checklist was used to gather data (included as, S1 File). The questions and contents of the tools were developed based on the World Health Organization tool to evaluate the quality of care delivered to sick children attending outpatient facilities [21] and the iCCM guideline developed by the Ethiopian Ministry of Health [22]. Each questionnaire was translated into three local languages (Afan Oromo, Amharic and Tigrigna) and thereafter back-translated by independent translators. The questionnaires in these languages were uploaded on tablets (CSPro 6.3) for data collection. The questionnaire was pilot tested and revised. The pilot test of questionnaires and procedures was done for three days in districts of Oromia and Amhara regions not included in the current study.

Sick children visiting a health post were assessed for eligibility. The consultations of sick children were performed by health extension workers who were on duty at health posts on the day of data collection. In case of two or more health extension workers in a health post, we included all as long as they saw a child eligible for inclusion. Observation of the clinical consultation was conducted by iCCM-trained data collectors who had received refresher training prior to the data collection. Data collectors were public health officers and nurses with a bachelor's degree in health science.

The overall process of the survey was monitored by trained supervisors who were also health professionals with a qualification of bachelor's degree and above. In addition, a data manager assigned at the central office provided daily data quality checks and feedback to the field teams.

## Data management and analysis

Data were monitored in the field by supervisors and checked by data managers for completeness, consistency and duplication of case records through visual scanning and running of frequencies.

The characteristics of the children were described. This was followed by a clinical pathway analysis. First, a description was done for the caregiver's complaints regarding the sick child. Second, the health extension worker's assessment and illness classification were analyzed. Finally, an analysis was done on the management and treatment provided to the child in relation to the classification of disease. For each step, i.e., assessment, classification, treatment, referral, and counselling, the findings were compared with the iCCM guidelines (Box 1). For key indicators we calculated proportions and 95% confidence intervals. All analysis was performed using SPSS version 20 (IBM corporation, New York, USA).

## Box 1. Definition of key terms.

Correct assessment: completeness of the assessment of the sick child by HEWs according to the iCCM clinical guidelines.

Classification: the presumptive illness assigned to the child after examination.

Correct classification: all classifications of sick children made by HEWs matched to the assessment result and iCCM clinical guidelines.

Correct treatment: all treatments of sick children made by HEWs matched to the classification results and iCCM clinical guidelines.

Quality of care: judged based on whether sick children correctly assessed, classified, treated and referred based on iCCM guidelines; whether caregivers were received comprehensive elements of counseling based on iCCM clinical guidelines.

### Ethical considerations

Ethical approval was obtained from the Ethiopian Public Health Institute (protocol number SERO-012-8-2016; 001 August 2016), The London School of Hygiene &Tropical Medicine (protocol number 11235, June 2016) and Jimma University (protocol number IHRPGD/472/2018, August 2018). Research permits were obtained from the Regional Health Bureaus in Amhara, Oromia, SNNP, and Tigray. Further, informed written consent was taken from caregivers of the sick children. Misdiagnosed children were provided correct treatment. Severely ill children were urgently referred to nearby health centers and, when needed, transportation was provided.

## Results

Of the planned 200 clusters, 6 were excluded for security reasons and data were gathered from 194 clusters. Twenty-five of the clusters had shared health posts and 22 of the health posts didn't have children mobilized with only 147 health posts in the clusters we included. Overall, 620 sick child consultations with health extension workers were observed across the 147 health posts; on average 4 sick children per health post. The mean consultation duration was 21 minutes (SD 11).

### Characteristics of sick children and health extension workers

A bit more than half of the sick children were boys (54%) and the majority was in the age group 2–23 months. As a consequence of the selection of study areas, most sick children were from Amhara region followed by Oromia region (Table 1). The accompanying caregivers were women (94%), most of them the biological mothers (92%).

Ninety-one percent of the sick child consultations were done by health extension workers, who had been trained in iCCM. Out of these, the majority (72%) had been trained before the past 12 months and only 19% of consultations were performed with health extension workers who had received training within the past 12 months. Over three-quarters (78%) of the sick children had been examined by health extension workers who had been supervised by health

**Table 1. Characteristics of sick children seen at health posts in four regions of Ethiopia.** December 2016-February 2017.

| Child characteristics | | Frequency | Proportion (95% confidence interval) |
|---|---|---|---|
| | | N = 620 | |
| Sex | Boys | 337 | 54 (50–58%) |
| | Girls | 283 | 46 (42–50%) |
| Age(months)* | 2–11 | 201 | 33 (29–36%) |
| | 12–23 | 190 | 31 (27–35%) |
| | 24–35 | 95 | 15 (13–18%) |
| | 36–47 | 75 | 12 (10–15%) |
| | 48–59 | 57 | 9 (7–12%) |
| Region | Amhara | 265 | 43 (39–47%) |
| | Oromia | 197 | 32 (28–36%) |
| | Tigray | 90 | 15 (12–18%) |
| | SNNP[a] | 68 | 11 (9–14%) |

*Missing data on two children (N = 618); [a]Southern Nations, Nationalities and Peoples' region.

center staff at least once in the past six months. Less than half (46%) were managed by health extension workers who had participated in a performance review and clinical mentoring meeting in the past six months. A majority (83%) of the sick children were managed by health extension workers who were residents of the local community, i.e., the Kebele, where the health post was located. The health extension workers utilized their chart booklets (92%) and iCCM registration books (93%) in the majority of their sick child consultations.

## Presenting complaints

A majority of the caregivers reported that their child had cough and related respiratory complaints (58%), followed by diarrhea (36%) and fever (26%). Caregivers often mentioned more than one complaint. Less frequent complaints included skin rash or skin infection (14 children, 2.3%), poor appetite (9 children, 1.5%), eye problems (8 children, 1.3%) or abdominal pain (5 children, 0.8%). There were also isolated complaints of burns, inability to stand, nasal discharge and redness of the umbilicus as a reason for the visit (Fig 1).

## Assessment and management

**Respiratory complaints.** There were 359 children with respiratory complaints (Fig 2). Half (50%, 95% CI 45–55) of the children with these complaints were assessed for history of the three general danger signs according to guidelines (ability to drink or breastfeed, vomiting everything, or convulsions). In addition, chest indrawing was assessed in 37% (95% CI 32–42) and stridor assessed in 25% (95% CI 21–30) of the children with respiratory complaints.

The health extension workers counted the respiratory rate for one minute in 74% (95% CI 69–78) of the cases with respiratory complaints. Of those assessed for respiratory rate, 33% (95% CI 28–39) had fast breathing. Further, among those found to have fast breathing, 78% (95% CI 69–86) were classified as suspected pneumonia by the health extension workers. Additionally, the health extension workers classified eight children with respiratory complaints whose respiratory count was not assessed as having pneumonia, as were 13 children who were considered to have normal breathing.

Overall, 47 of the 90 children classified to have suspected pneumonia (52%, 95% CI 42–62) received antibiotics (amoxicillin). The health extension workers provided antibiotics to 34 out of the 269 children (13%, 95% CI 9–17) with respiratory complaints who were found not to

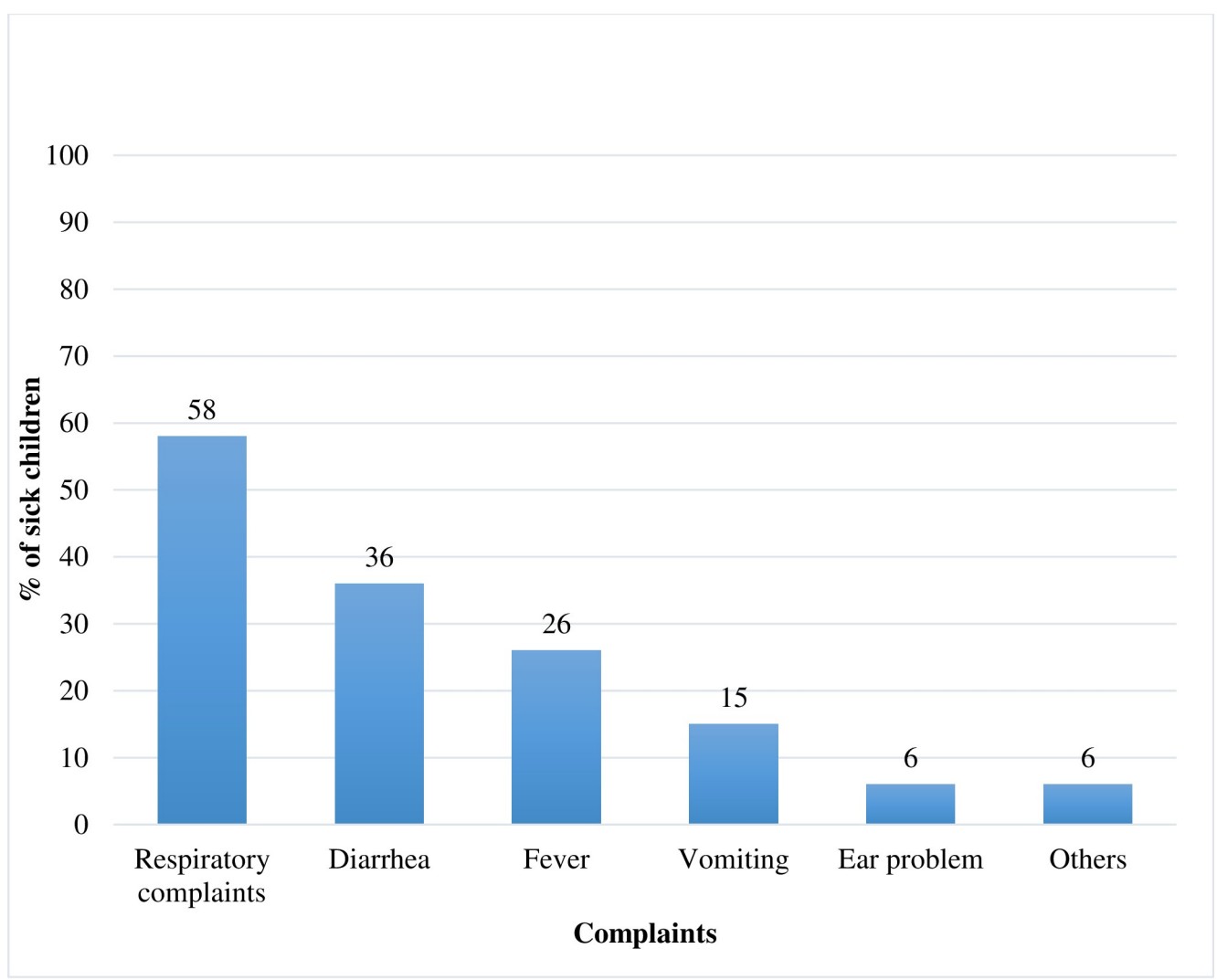

**Fig 1. Complaints of sick children seen at health posts of four regions of Ethiopia.** December 2016-February 2017.

have pneumonia. A closer look at the sick children who were labeled to have fast breathing and hence suspected pneumonia revealed that 57% (95% CI 46–67) of these children were treated with antibiotics.

**Diarrhea complaints.** There were 224 children with diarrhea complaint (Fig 3). The health extension workers assessed nearly six in ten (59%, 95% CI 53–66) of the cases for the guideline's three general danger signs (ability to drink or breastfeed, vomiting everything or convulsions). The most frequently assessed general danger signs were the ability to drink or breastfeed (84%) and vomiting everything (80%). The health extension workers assessed convulsions in 67% of the sick children. Moreover, six out of the ten sick children who were labelled as not visibly awake by the health extension workers, were assessed for lethargy or unconsciousness.

The health extension workers examined the presence of dehydration in 33% (95% CI 27–40) of the 210 sick children labeled to have diarrhea. Out of these, 13% were classified as being dehydrated. Additionally, among the 140 children with diarrhea who were not assessed for

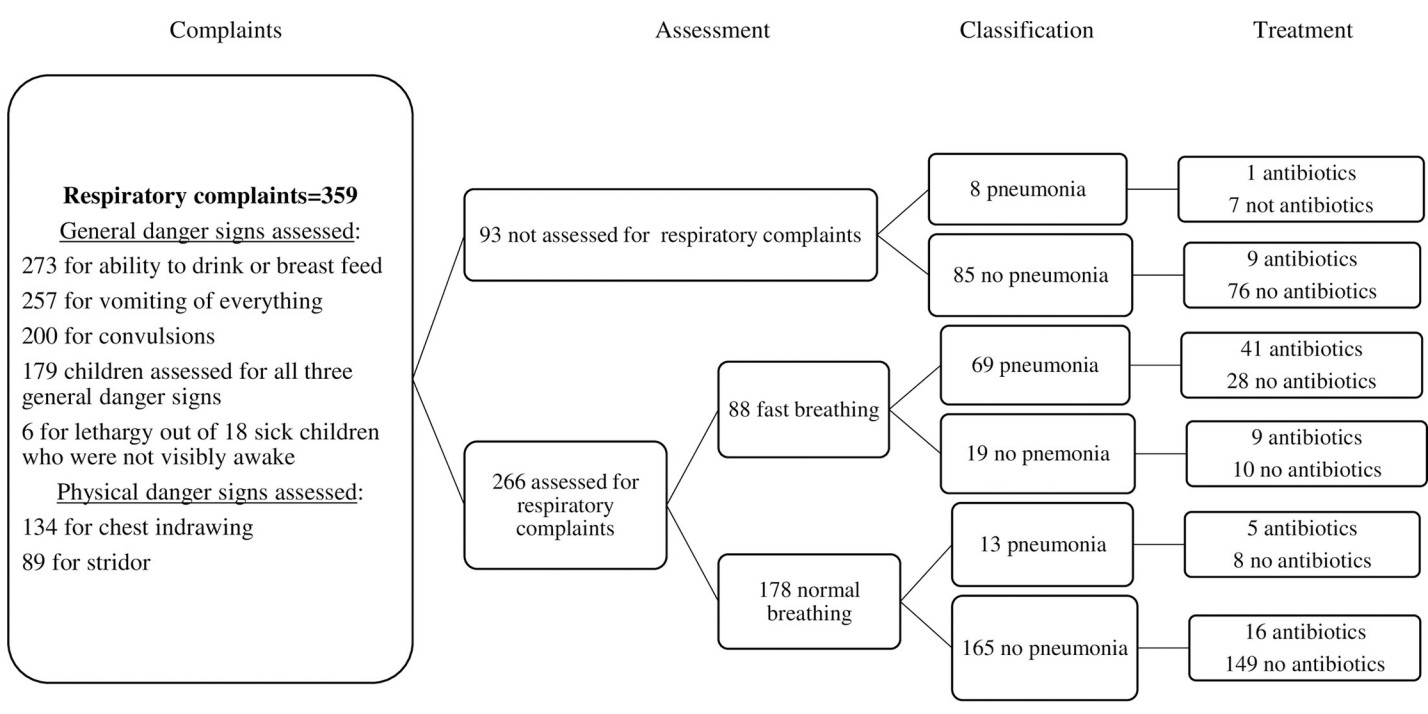

**Fig 2. Clinical pathway analysis of cases with respiratory complaints.**

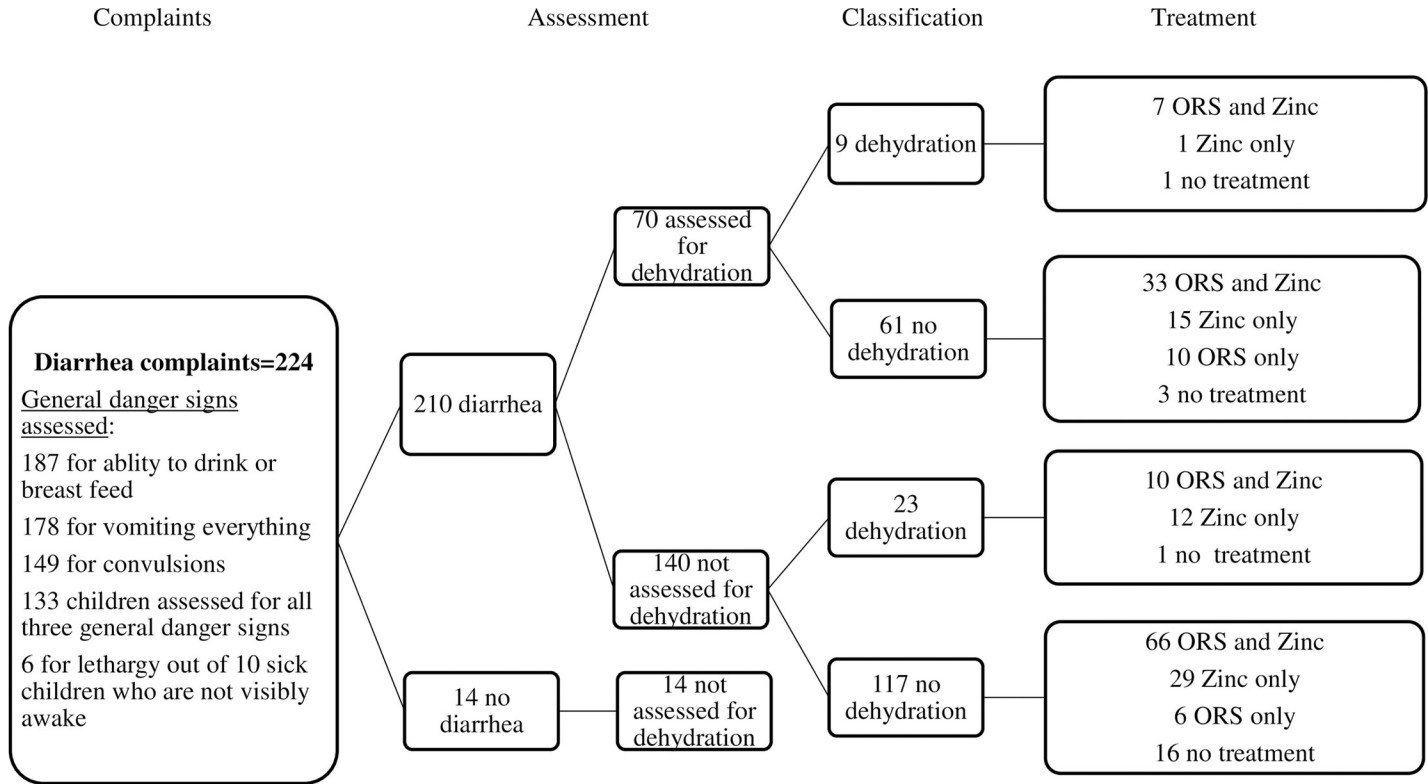

**Fig 3. Clinical pathway analysis of cases with diarrhea as a chief compliant.**

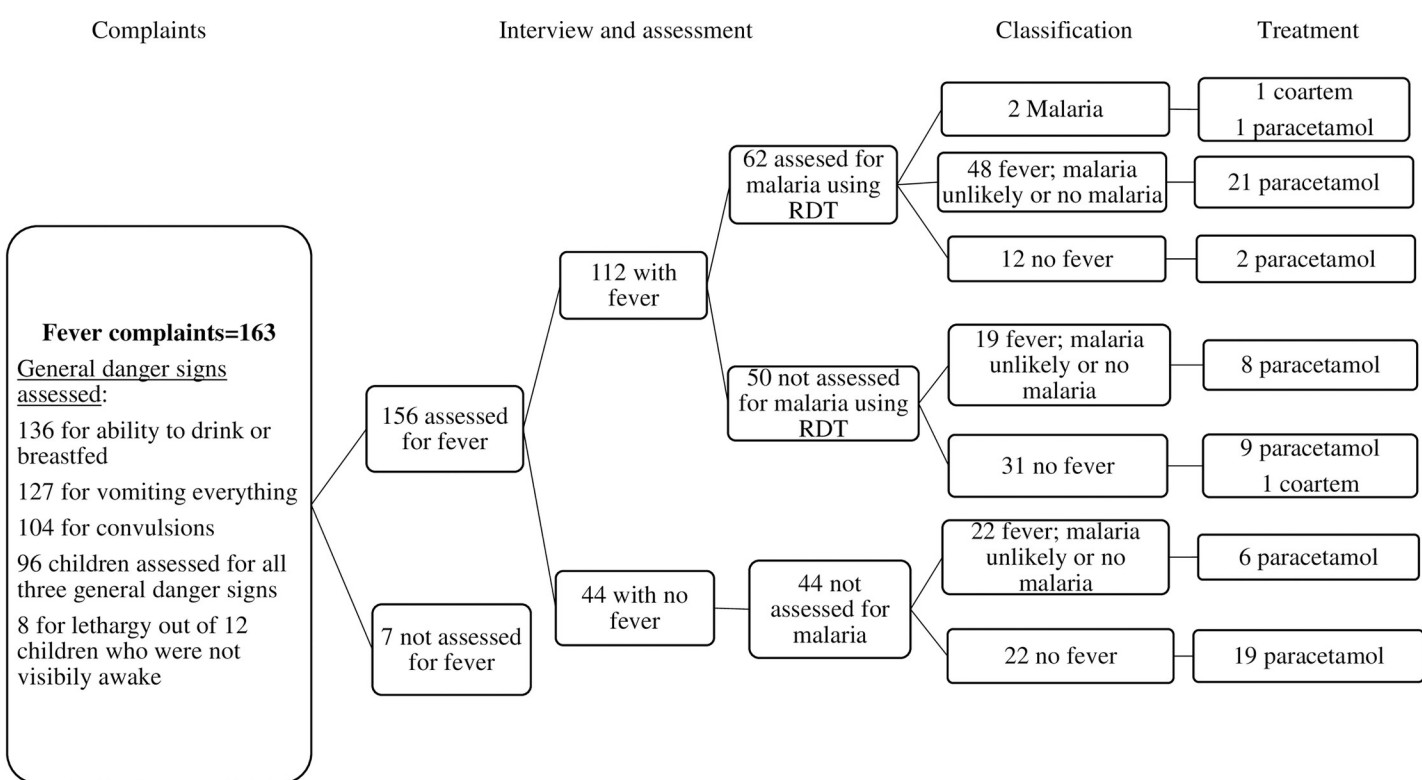

**Fig 4. Clinical pathway analysis of cases with fever as a chief complaint.**

dehydration, the health extension workers classified 16% (95% CI 11–23) as having dehydration.

The health extension workers provided oral rehydration solution (ORS) treatment to 53% (95% CI 36–70) of children labelled to have diarrhea with dehydration and zinc treatment to 94% (95% CI 81–99) of these cases. Both ORS and zinc treatments, which the guidelines prescribe, were provided only to over half (53%, 95% CI 36–70) of children labelled to have diarrhea with dehydration. The health extension workers also provided ORS (65%, 95% CI 57–71) and zinc treatments (80%, 95% CI 74–86) to sick children who were labelled as having diarrhea without dehydration.

**Fever complaints.** There were 163 children with fever complaints (Fig 4). The health extension workers assessed nearly six in ten (59%, 95% CI: 51–66) of these children for the guideline's three general danger signs (ability to drink or breastfeed, vomiting everything and convulsions). The most assessed general danger signs were ability to drink or breastfeed (83%) and vomiting everything (78%).

Health extension workers assessed fever in 96% (95% CI 92–98) of sick children with complaints of fever by using thermometer. Out of these, 72% (95% CI 64–78) were labelled to have fever. Out of those labelled as having fever, the health extension workers followed the guideline and tested malaria by using rapid diagnostic test (RDT) in slightly over half (55%, 95% CI 46–64) of the children. Additionally, health extension workers also examined meningitis in 13% and measles in 24% of sick children labelled as having a fever.

Out of the 62 children examined for malaria, 2 were RDT positive and classified as having malaria, 48 as fever with unlikely or no malaria, and 12 as no fever. In addition, health extension workers also classified some children who had a high temperature as having fever with

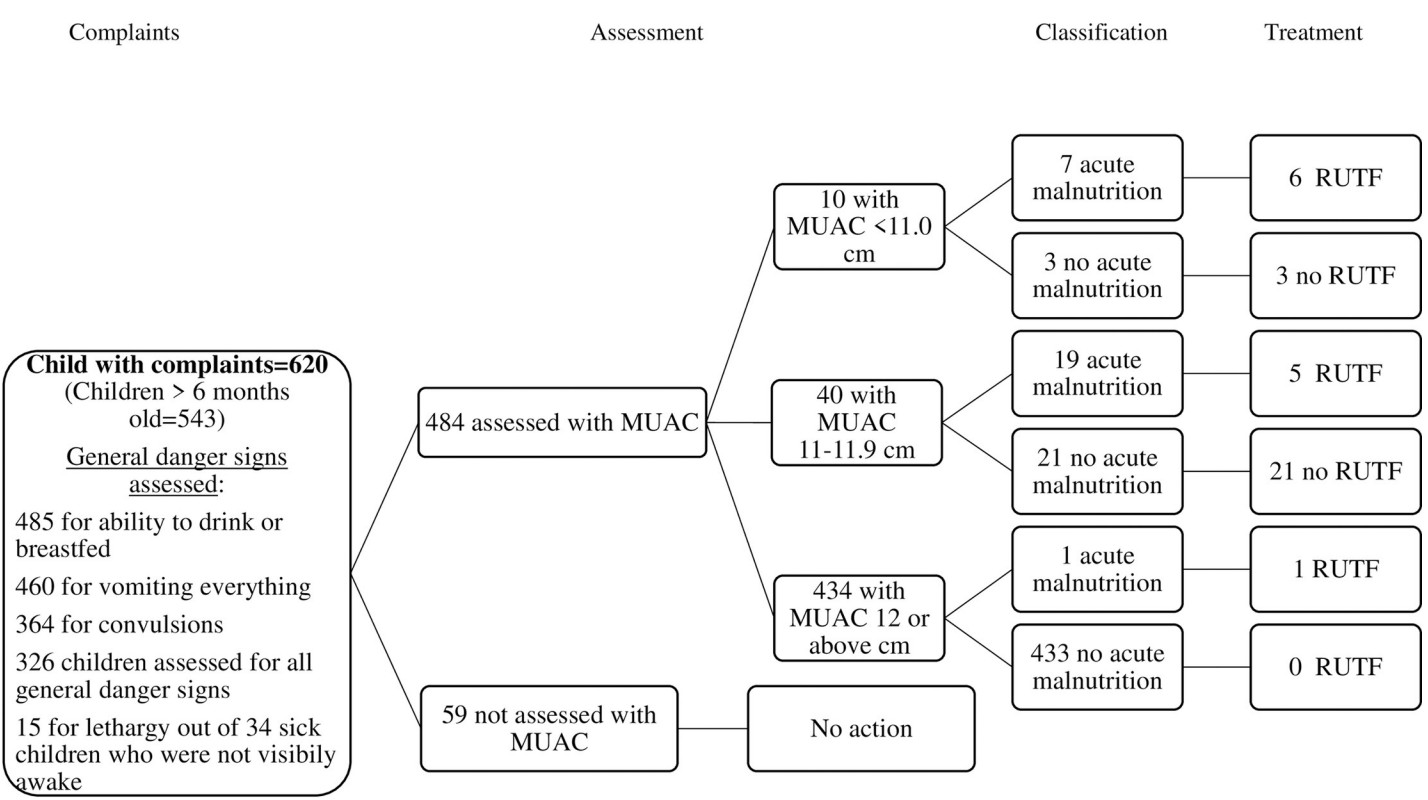

**Fig 5. Clinical pathway analysis of all cases for checking presence of malnutrition.**

unlikely or no malaria, without performing an RDT assessment. Similarly, among those that had normal temperature, some were classified as having fever.

Coartem treatment was provided to one of the two children classified as having malaria. Additionally, one child who was labelled as having a fever by the health extension worker but not examined for malaria was provided coartem treatment. Overall, the health extension workers provided paracetamol treatment to 37% (95% CI 28–46) of children with fever and 57% (95% CI 42–71) of the children labelled as no fever.

**Ear problem complaints.** The health extension workers also assessed and managed children with ear problem (S1 Fig). Among those complaining of an ear problem, the health extension workers assessed the three general danger signs in 38% (95% CI 23–54). Twenty-two out of these children (63%) were classified to have an ear infection (acute or chronic) and 10 were treated with antibiotics.

**Malnutrition.** The national iCCM guideline recommends that health extension workers should assess all children for malnutrition, either well or sick, by measuring the mid-upper arm circumference (MUAC). Nearly nine in ten of the sick children were assessed for malnutrition by MUAC measurement (89%, 95% CI 86–92). Ten percent or 50 children had a circumference less than 12.0 cm, which is the Ethiopian cut-off for acute malnutrition. The health extension workers classified only half (52%, 95% CI 38–66) of these 50 cases as having malnutrition and 11 (22%, 95% CI 12–35) were treated with Ready-to-Use Therapeutic Food (Fig 5).

The health extension workers assessed the three general danger signs in half (53%, 95% CI 49–57) of the 620 sick children who visited the health posts with different complaints (Fig 5). The health extension workers assessed visible severe wasting in a bit less than five in ten

children (47%, 95% CI 36–58) and bipedal edema in slightly over a third (37%, 95% CI 33–41) of the children. Vaccination status was assessed in 73% (95% CI 70–77) of the children and vitamin A supplementation status (59%, 95% CI 55–63) and anemia (51%, 95% CI 47–55) were assessed in slightly over half of the children (S1 Table).

Overall, the health extension workers provided classifications for 264 children with an illnesses and out of these, 52% (95% CI 46–58) were provided with treatment.

**Referral.** The health extension workers concluded that 55 children had one or more of the general danger signs, which required urgent referral. Referral was, however, recommended for only 11 (20%, 95% CI 11–32) of these children. In ten of these referrals, the health extension workers explained the need for referral and eight of the referrals were accepted by the caregivers. The health extension workers wrote a referral note for five of the referrals and they had arranged transportation for one of the referrals.

Of the 87 children, who were labeled to have fast breathing by the health extension workers, ten were referred. The major reasons for these referrals were severe illness and drug stock out. The health extension workers explained the reason for referral to the majority of the caregivers, although only six in ten received a referral note. The health extension workers referred one-third of sick children with uncomplicated severe malnutrition (MUAC <11 cm) who could have been treated at the health post. The main reasons for these referrals were severe illnesses and drug stock out. The health extension workers explained and provided referral notes to all of the referred cases with uncomplicated severe malnutrition.

**Counseling.** A majority of the caregivers were counseled on home-based care (89%) and continued child feeding (85%). The health extension workers counseled on increased fluid intake to 18% of the caregivers. Comprehensive counseling (including all recommended components) was provided to slightly more than one third of the caregivers (Table 2).

## Discussion

We have shown that sick children presenting at health posts in four regions of Ethiopia were to a large extent not assessed, classified, managed and treated according to the clinical guidelines for the integrated Community Case Management of childhood illnesses. Almost half of the sick children were not assessed for general danger signs. Three-quarters of the sick children presenting with respiratory complaints were assessed with counting of respiratory rate. Few children with diarrhea were assessed for dehydration. The health extension workers usually measured arm circumference for malnutrition screening. The results of the assessments were

**Table 2. Proportion of caregivers who received counseling at health posts of four regions of Ethiopia.** December 2016 to February 2017.

| Components of counseling | N | % | [95% confidence interval] |
|---|---|---|---|
| Health extension workers advised: | | | |
| On home-based care | 613 | 89 | (86–91%) |
| To return if a child can't drink or breast feed | 616 | 80 | (77–83%) |
| To return if a child gets worse | 616 | 82 | (79–85%) |
| Caregiver to increase fluids | 617 | 18 | (15–21%) |
| Caregiver to continue feeding | 619 | 85 | (82–87%) |
| To continue breastfeeding or breastfeed more frequently | 619 | 77 | (74–80%) |
| On when to return for follow-up | 619 | 82 | (79–85%) |
| *Health extension workers offered comprehensive packages of counseling | 612 | 38 | (35–42%) |

*Comprehensive including all seven elements of counseling.

to a large extent not logically influencing illness classification, management, and treatment. The clinical pathway analysis indicated that this lack of adherence to the guidelines led to mis-diagnoses and failure to provide potentially life-saving treatments.

The health extension workers assessed three quarters of the sick children with cough by counting their respiratory rate but failed to correctly classify and treat those that should be managed as suspected pneumonia according to the guidelines [22]. In another sub-study of our larger baseline survey, the sick children included in this study were re-examined by a clinical officer immediately after the observed examination and management session reported here. Only 60% of children with acute respiratory tract infection (suspected pneumonia) were correctly identified (sensitivity) by the health extension worker [17]. Antibiotic treatment was provided to a proportion of the children classified as suspected pneumonia and also to some children without any suspected pneumonia. Such actions might lead to under- as well as over-use of antibiotics and poor targeting of antibiotic treatment. An earlier Ethiopian study of iCCM quality-of-care was performed in one of the regions in 2012 –one year after the introduction and training in this management program. At that time, 93% of sick children with cough and respiratory problems had their respiratory frequency assessed and 72% of the suspected pneumonia cases were judged to be correctly treated [23]. The current study was performed when the management program was fully scaled-up and integrated into the routine services at the primary care level. A study done with community health workers in Malawi found similar levels of counting of respiratory rates in children with respiratory problems [14] while a report based on recently trained community health workers in Tanzania [24] had a more optimal assessment of sick children with cough (91%). At the end of the Millennium Development Goal era in 2015, half of the world's deaths due to pneumonia occurred in five countries, i.e., India, Nigeria, Pakistan, Democratic Republic of the Congo, and Ethiopia [25]. In order to continue reducing these deaths and reach the Sustainable Development Goals for child mortality, improved quality of the primary health care services for children with suspected pneumonia is needed.

Many children suffered from diarrhea, but only one third were assessed for dehydration. The health extension workers classified children as dehydrated or not, partly disregarding whether this had been assessed, and the treatment with ORS and zinc was not consistent with the assessment and classification. These practices were not aligned with the treatment guidelines for children with diarrhea and dehydration [22] and may potentially lead to poor outcomes. In the earlier mentioned Ethiopian study one year after the introduction of the integrated Community Case Management program, the assessment and management were more appropriate [23]. Also the earlier mentioned studies in Malawi and Tanzania reported better alignment with guidelines in the classification and management of diarrhea [14, 24].

Almost all sick children with fever complaints had their temperature taken, but the health extension workers tested the presence of malaria with rapid diagnostic tests in only a fraction. This finding should be seen against the fact that most study districts were located in highland areas and the study was conducted in a low-transmission season.

All children from six months of age should, according to the guidelines, be assessed for malnutrition [22]. The health extension workers assessed malnutrition in nine of ten children by mid-upper arm circumference. The measurements were, however, not consistently used to classify children who suffered from severe or moderate acute malnutrition. Further, treatment with Ready-to-Use Therapeutic Food was not provided according to the measurements or classification. In the sub-study reporting disease classification also by re-examination of the same sick children as in this study, only 40% were correctly classified for malnutrition [17]. The physical danger signs, i.e., visible severe wasting and pitting edema, were also assessed in few children.

In another report based on the same large survey as this study, referral practices were weak at all levels within the Ethiopian primary care system [26]. We found that less than one-fifth of the children who, according to guidelines, needed referral were referred to a higher-level health facility. The need for referral was explained to most of the caregivers, but referral notes were rarely provided. Lack of referral notes might have affected the continuity of care as information was not forwarded [27, 28]. Non-adherence to referral guidelines was also observed among community health workers in Uganda [29] and Ghana [30] while a reasonable number of referrals supported with referral slips was observed among community health workers in Pakistan [27]. Most caregivers were advised on home-based sick child care and continued feeding. This was better counselling than findings in the previously mentioned study in Tanzania [24].

The performance of the health extension workers is shaped by various factors. Training in clinical management skills as well as the supply of equipment and pharmaceutical drugs are prerequisites for good quality services. Supportive supervision, performance reviews, and clinical mentorship may improve the quality of care provided [31–33]. In our study most health extension workers had been trained on the clinical management of sick children, but only a fifth had been trained in the past 12 months. Further, the regular supportive supervision from health centers and the clinical mentoring were often missing. This implies that even if most health extension workers originally had been trained in managing sick children, the continuous re-training and support were frequently missing. The quality of clinical training, mentorship, supervision and review meetings could influence impact. Moreover, the health extension worker's relationships with the community, other contextual factors, and the wide range of tasks given to these workers can hugely affects their performance [34–45].

To the best of our knowledge, this is the first study analyzing the whole process of being assessed, classified, managed, and treated for individual sick children 2–59 months of age at the Ethiopian primary care level seeking services provided by the integrated Community Case Management of common childhood illnesses. The study was conducted in 52 districts of the four main agrarian regions of Ethiopia as part of a baseline survey for a demand-creation intervention that aimed at increasing the utilization of primary child health services. The sample was not randomly selected to represent these regions, but we have no reason to believe that the results differ from the average situation in rural parts of these four regions. The participating sick children were mobilized to attend the health posts at the day of study. This could potentially lead to a biased sample with more or less severe cases than an average day at the health post. The mix of complaints were, however, the expected proportions of respiratory complaints, diarrhea, fever, as well as malnutrition [7]. Observations of the sick child consultations were conducted by trained health officers and nurses, who had received refresher training on iCCM. The observation method could be influenced by the Hawthorne effect, i.e., a change in professional behavior of the health extension workers when being observed, so that they could be more than usually careful in their work [46, 47]. However, the findings showed that health extension workers were not performing well in all components of care. This implies, that the normal management situation could suffer even more from lack of adherence to treatment guidelines and procedures.

In summary, this study has showed major quality gaps in the assessment, classification, and provision of curative care for sick under-five children at health posts. The Ethiopian health extension workers' clinical assessment, classification and management of sick children did, to a large extent, not follow the clinical guidelines. This lack of adherence could lead to misdiagnoses of important causes of mortality, such as suspected pneumonia, diarrhea with dehydration, and acute malnutrition. These gaps display obvious risks of sick under-five Ethiopian children not receiving potentially life-saving treatments.

The findings in our study have practical implications for policy and practice. First, the quality of care gap observed at the health post level should alert policy makers to seriously look into the competence of the health extension workers in providing clinical care for sick children. This, of course, requires looking into findings from other similar evaluation reports targeted at examining the health extension program in general and the performance of the health extension workers in particular. If these frontline workers should continue treating sick children, our findings call for serious attention to the monitoring and appraisal of their clinical care practices. Furthermore, these providers need high-quality and continuous support with practical in-service training, mentorship and adequate supply of medicine and equipment.

## Supporting information

**S1 Fig. Clinical pathway analysis of cases with ear problem as chief complaint.**
(TIF)

**S1 Table. Proportion of sick children assessed for physical danger signs, vaccination status and vitamin A status by complaints category.**
(DOCX)

**S1 Box. Health Service Package for Health Service Extension Program.**
(DOCX)

**S1 File. Observation checklist.**
(ZIP)

## Acknowledgments

Our sincere gratitude goes to all participants in the study and to regional and district-level health system administrations for their support to the study.

## Author Contributions

**Conceptualization:** Dawit Wolde Daka, Muluemebet Abera Wordofa, Mirkuzie Woldie, Lars Åke Persson, Della Berhanu.

**Data curation:** Dawit Wolde Daka, Lars Åke Persson.

**Formal analysis:** Dawit Wolde Daka, Lars Åke Persson.

**Funding acquisition:** Muluemebet Abera Wordofa, Della Berhanu.

**Investigation:** Dawit Wolde Daka, Muluemebet Abera Wordofa, Mirkuzie Woldie, Lars Åke Persson, Della Berhanu.

**Methodology:** Dawit Wolde Daka, Muluemebet Abera Wordofa, Mirkuzie Woldie, Lars Åke Persson, Della Berhanu.

**Project administration:** Dawit Wolde Daka, Mirkuzie Woldie, Della Berhanu.

**Supervision:** Dawit Wolde Daka, Muluemebet Abera Wordofa, Mirkuzie Woldie, Lars Åke Persson, Della Berhanu.

**Validation:** Dawit Wolde Daka, Muluemebet Abera Wordofa, Mirkuzie Woldie, Lars Åke Persson, Della Berhanu.

**Visualization:** Dawit Wolde Daka, Lars Åke Persson, Della Berhanu.

**Writing – original draft:** Dawit Wolde Daka.

**Writing – review & editing:** Dawit Wolde Daka, Muluemebet Abera Wordofa, Mirkuzie Woldie, Lars Åke Persson, Della Berhanu.

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
