## [Decision Letter · Decision Letter 0]

4 Jun 2020

PONE-D-20-06207

Quality of clinical assessment and management of sick children by Health Extension Workers in four regions of Ethiopia: a cross-sectional survey

PLOS ONE

Dear Dr. Daka,

Thank you for submitting your manuscript to PLOS ONE. After careful consideration, we feel that it has merit but does not fully meet PLOS ONE’s publication criteria as it currently stands. Therefore, we invite you to submit a revised version of the manuscript that addresses the points raised during the review process.

We look forward to receiving your revised manuscript.

Kind regards,

Elizeus Rutebemberwa

Academic Editor

PLOS ONE

Journal Requirements:

2. Please include additional information regarding the survey or questionnaire used in the study and ensure that you have provided sufficient details that others could replicate the analyses. For instance, if you developed a questionnaire as part of this study and it is not under a copyright more restrictive than CC-BY, please include a copy, in both the original language and English, as Supporting Information. In addition, please provide details of the pilot testing of this questionnaire, including the number of participants and where they were recruited from.

Reviewers' comments:

Reviewer's Responses to Questions

**Comments to the Author**

1. Is the manuscript technically sound, and do the data support the conclusions?

Reviewer #1: Yes

Reviewer #2: Yes

2. Has the statistical analysis been performed appropriately and rigorously? 

Reviewer #1: Yes

Reviewer #2: I Don't Know

3. Have the authors made all data underlying the findings in their manuscript fully available?

Reviewer #1: Yes

Reviewer #2: Yes

4. Is the manuscript presented in an intelligible fashion and written in standard English?

Reviewer #1: Yes

Reviewer #2: No

5. Review Comments to the Author

Reviewer #1: Overall comments

1. This is an excellent and important, needed study. I recommend its publication and suggest only minor revisions.

2. I think the conclusion in lines 452-455 is a dangerous overstatement. Of course, taking away the management role from HEWs is a theoretical possibility and having them refer all sick children to a higher facility with higher-level workers. But is this feasible, especially in Ethiopia where the next level of care is a 3-4-hour walk away and transport is not available? And what is the evidence that higher-level workers trained in iCCM perform better? I am not opposed to keeping this in the paper. However, the real message – at least from my standpoint – is to strengthen supervision and retraining. I think it would be important to investigate whether the supervisors have had training in iCCM and how well they supervise HEW performance in this area. Maybe the MOH should consider having a special cadre of iCCM supervisors who do nothing but provide ongoing training and supervision on this one aspect of the HEWs’ work.

3. There is a need to better document to contribution of stockouts to lack of adherence to guidelines. Of course, better logistical support is an obvious need as well.

Questions/issues that I think should be addressed

1. How were the 52 districts selected?

2. On p. 6, lines 151-2: malnutrition, anemia, HIV and immunization/vitamin A status are not symptoms. How anemia and HIV status are determined might be explained. Malnutrition is assessed by MUAC, of course, and immunization/vit A status were presumable assessed by asking the caretaker.

3. How was the community mobilization to get children to attend the health posts carried out?

4. It would be important to document if possible when the iCCM training was provided and what kind of in-service refresher training had been provided. It would also be important to know if the supervisors of the HEWs had been trained in iCCM since then? The findings suggest that this might be one practical implication of the study findings – along with more frequent in-service training on iCCM or perhaps have HEWs go through the entire training again from time to time. After all, recertification is a common requirement for many trainings in the US (cardiopulmonary resuscitation, specialty boards, etc.). What is the evidence from other studies about the “decay rate” in knowledge and practices for CHWs after iCCM? Is there any?

5. How often were drug stockouts present?

6. What are the criteria for RDT testing? Are these contextually appropriate? Should all children with fever have had an RDT for malaria?

7. Were Ready-to-Use Therapeutic Foods actually available at the health post?

8. Don’t HEWs also treat sick children out in the home and in the community? You might at least want to mention that.

9. The paper should at least mention all of the various tasks and roles that HEWs perform beyond iCCM.

Minor suggestions for revisions

1. The “v” in vitamin A (line 122, p. 6) should not be capitalized

2. The designation for HEWs is not consistent through the paper: sometimes it is “Health extension worker,” sometimes “health extensions worker,” sometimes “HEW.”

3. Did the observer also count the respiratory rate? How accurately did HEWs accurately count AND CLASSIFY the respiratory rate?

Additional minor comments

1. It would be interesting to know if the 9% of HEWs observed who had not received training in iCCM performed as well as those who had been.

2. An interesting follow-up study would be to talk with CHWs about why they did not follow the iCCM guidelines – was it truly a lack of knowledge or other not readily apparent (to the researcher) other complicating factor?

3. P. 20, line 442, should be “workers’ ” not “workers”

Reviewer #2: • The authors have written a paper which provides an important perspective on understanding the quality of care provided by health extension workers, and considerations for improving the care quality gap. While these results could add to the body of literature on understanding and strengthening quality in hard-to-reach settings, it requires revisions and clarification of the relationship to the broader study and how this was related to an earlier publication which compared technical quality with a re-exam of the patient. The submission needs changes to improve the clarity of writing and particularly in the methods, and results and discussion detailed below. The manuscript also needs a careful review for grammar and formatting and would consider a copy-editor before resubmission; the manuscript contains multiple typos (for example, lines 62, 198), or unclear or wordy sentences (e.g., 114; 128; 135 – inconsistent capitalization/acronym usage for HEW

Major and minor comments.

• It is unclear how these findings fit into the overall "Optimizing Health Extension Program" interventions (those which you've listed do not seem directly related to this project), and the trial registration number you've given does not mention OHEP.

• The relationship to this study and published results also needs clarification. You refer to another "recent study, based on the same larger survey as this article, [which] showed that the health extension workers’ diagnostic accuracy was low" (ref 17) which included gold standard as re-exam, yet this paper only described observation with a checklist. Why were data from the re-exam not used? For assessments, what were the differences between observation and re-exam?

• This manuscript as "part of a larger baseline" survey, but you later say that you calculated power to "correctly assess, classify and treat diseases at baseline vs endline surveys, in intervention and comparison areas." (line 147) and use of a re-exam. Please clarify

• Minor: You state that "no previous study has analyzed the entire process of managing sick children at Ethiopian health posts" (line 95), but this is only related to IMCI and children 2months-5 years. This sentence should include these caveats. on describing the symptoms most commonly presented to health posts.

Methods

• The selection and recruitment processes are unclear and should be consolidated. In line 127, you state "participants were invited," but earlier you state that children who showed up at the health facility were included. The paper later states that community mobilization targeted sick children to have them brought to the facility – which children were targeted and why, and how did community mobilization work? You also state that children "brought to the health post by their caregivers and the health extension workers were eligible" (137) – did HEWs select children for the study in the community and then bring them to the facility?

• In line 140, you aimed for an average of 4 sick children per health post – was that a target, or a cap, or a minimum, and what efforts were made to include 4 children? How were the children included selected (who determined they had IMCI-eligible symptoms). Were extremely sick children excluded to not delay referral if needed with the re-exam? The statement "this implies that the sample was not randomly selected to represent the regions"? (359-360). Explanation for the lower number included (620 versus 800) should be explained and if entire health posts were not included (beyond ones excluded for security). Were the final regional distribution of children (Table 1) as planned?

• How were HEWs selected? If there are two HEWs at the health facility, who conducted the consultation and how was that decided? Were any HEWs excluded for any reason?

• You report that data were collected through a structured observation checklist on tablets (160), but earlier state that "re-examiners" (146) were considered the gold standard. Did re-examination occur? If not, how did you determine whether there was a misdiagnosis? (181) The checklist should be included as well as how they determined if specific actions such as counting breaths were determined. If a child had more than one eligible complaint, were both included?

• Who are the trained supervisors (166)? What was their role and how was it different from the data collectors?

• What kind of data quality checks occurred and what were the results?

• The description of power calculation is unclear (142-147) – it states that the study was powered to "correctly assess, classify and treat diseases at baseline vs endline surveys, in intervention and comparison areas" (148) –. In line 149, you then state that the "study was powered to "ascertain prevalence (for example when performing a certain assessment)" – prevalence of what?. Finally, if only 620 children ultimately included how did the power increased from 80% to 85%?

Ethics

• It is important to note if there any considerations for delay of care for severely ill children if gold standard re-review was included (see questions above)

• How was consent gathered form non-biological parents? (187)

• Were children who required urgent referral but were not referred by HEWs referred by the supervisor or data collector (312)?

•

Results and Discussion

• Throughout, please make it clear what is the standard of care. For example – are HEWs expected to ask about all three general danger signs (line 224), assess for chest-indrawing (225), examine for meningitis in fever cases (273) and when are they supposed to prescribe antibiotics for all cases? (237, 291)? The comparison to protocols is done better in the MUAC section (293), where you describe the expectations for HEWs as compared to their performance, although you do not mention whether correctness of MUAC procedure is assessed. While many readers may be familiar with IMCI, they may differ by country. Consider a table as a supplement

• I was confused by the data on fast breathing. For example, in line 228-231, You report that HEWs counted fast breathing in 74% of cases and only correctly labeled fast breathing in 78% of cases with fast breathing. How was that confirmed Especially if the HEW did not perform the exam?

• Line 201: you note that only 91% of sick child consultations were done by HEWs – who conducted the other 9%? If the study was to evaluate HEWs were these excluded? Same question for non-IMCI conditions (ex. Burns)

• Line 215: please clarify how were symptoms assessed? It is unclear if it is from caregiver report, examination, re-examination, or some other data collector assessment. For example – in line 215, was fever just based on complaint, evaluation by the HEW, use of thermometer, etc.?

• In limitations, I was not sure that I understand the statement line 360 you can have “no reason to believe” children represented the rural areas in the 4 regions. Can the authors add in a reference for example about the range of conditions and severity being “as expected”. Also typically limitations are towards the end of the discussions, not in the middle

• Understanding variability across regions would be important and consider as well a metric of overall quality per HEWs (for those with >1 observation-were there higher and lower performers or did they all provide similar overall quality (or challenges)?

• The discussion brings in data from other studies. A brief discussion why some results differ in this study is important. In addition, please include actual numbers (line 393 described the Tanzania results as more optimal.

• I was confused by the statement line 389 that the study was “performed when the management program was fully scaled up”-the timing of this assessment and if baseline or endline needs clarification.

• The statement starting line 440 was confusing and needs clarification

Tables and Figures

• While I appreciate the details in the figures flow diagrams, it is hard to look across tasks and diseases and without %, hard to determine rates. It would be better to present tables of the results of HEW performance in addition to the table on counseling, as the reader gets a bit lost in the text. Table 1 could be described an in a supplement if you need more space.

Minor comments

• 25 – In your abstract, you state that low care seeking may be due to unfavorable community perceptions of quality of care provided, but you do not support with references or other with evidence in the manuscript.

• 68 – Please clarify whether 37,000 HEWs have been deployed over the history of the program, or are 27,000 HEWs currently deployed?

• 249 – How is labelling by an HEW as not visibly awake different than assessing for unconsciousness?

• 233-235 – Include percent along with number of children.

• 251 –What does "examined the presence of dehydration" mean – is that through physical exam? Was the appropriateness/accuracy of the exam evaluated?

• 365 – How do you know what the expected mix of complaints would be?

6. PLOS authors have the option to publish the peer review history of their article (what does this mean?). If published, this will include your full peer review and any attached files.

Reviewer #1: Yes: Henry B. Perry

Reviewer #2: No

---

## [Author Response · Author response to Decision Letter 0]

4 Aug 2020

27 July, 2020

Editor,

PLOS ONE

Dear Editor,

We are grateful for the opportunity to submit a revised version of our manuscript “Quality of clinical assessment and management of sick children by Health Extension Workers in four regions of Ethiopia: a cross-sectional survey”. Below follows a point-by-point response to each of the comments and queries from the Editor and the reviewers. We refer to the lines in the manuscript where the changes are made. The changes are shown with track changes in the manuscript. We hope you are satisfied with this new version of the manuscript.

With best regards 

For the group of authors

Dawit Wolde Daka

Corresponding Author

1. Editor’s comment: Please ensure that your manuscript meets PLOS ONE's style requirements, including those for file naming. The PLOS ONE style templates can be found at:

Response: We have revised and checked that the entire manuscript and the labelling of files are prepared according to the PLOS ONE requirements. 

2. Editor’s comment: Please include additional information regarding the survey or questionnaire used in the study and ensure that you have provided sufficient details that others could replicate the analyses. For instance, if you developed a questionnaire as part of this study and it is not under a copyright more restrictive than CC-BY, please include a copy, in both the original language and English, as Supporting Information. In addition, please provide details of the pilot testing of this questionnaire, including the number of participants and where they were recruited from.

Response: We have added more details to the narrative (Line # 158 and following). The questionnaire was developed based on a WHO tool used to evaluate the quality of care delivered to sick children attending outpatient facilities and the integrated Community Case Management guideline of the Ethiopian Ministry of Health. The sources are cited in the manuscript. We translated the questionnaire to local languages (‘Afan Oromia, Amharic and Tigrigna’) by experienced translators and back-translated into English with independent translators. The questionnaire and procedures were pilot tested during three days in non-study districts of Oromia and Amhara regions as described in the revised version of the manuscript. As requested, we include the questionnaires as supplementary files.

3. Editor’s comment: We note that you have indicated that data from this study are available upon request. PLOS only allows data to be available upon request if there are legal or ethical restrictions on sharing data publicly. For information on unacceptable data access restrictions, please see http://journals.plos.org/plosone/s/data-availability#loc-unacceptable-data-access-restrictions.

Response: Data from this study are co-owned by the participating institutions and stored in a depository at the Ethiopian Public Health Institute (EPHI). The use of these data is guided by a data sharing agreement that states that data will be made available upon reasonable request but are not publicly available during the period when Ph.D. students and other involved researchers are analyzing and reporting based on these data. Data can be accessed from the secretary of ‘Data sharing committee of EPHI-LSHTM Collaborative Projects’.

Contact information:

Name: Mrs. Martha Zeweldemariam

E-mail: martha.zeweldemariam@lshtm.ac.uk

Reviewer #1: Overall comments

1. Reviewers comment: This is an excellent and important, needed study. I recommend its publication and suggest only minor revisions.

Response: We would like to thank you for your positive recommendation. 

2. Reviewers comment: I think the conclusion in lines 452-455 is a dangerous overstatement. Of course, taking away the management role from HEWs is a theoretical possibility and having them refer all sick children to a higher facility with higher-level workers. But is this feasible, especially in Ethiopia where the next level of care is a 3-4-hour walk away and transport is not available? And what is the evidence that higher-level workers trained in iCCM perform better? I am not opposed to keeping this in the paper. However, the real message – at least from my standpoint – is to strengthen supervision and retraining. I think it would be important to investigate whether the supervisors have had training in iCCM and how well they supervise HEW performance in this area. Maybe the MOH should consider having a special cadre of iCCM supervisors who do nothing but provide ongoing training and supervision on this one aspect of the HEWs’ work.

Response: Thank you for this comment. We agree that the previous phrasing was an overstatement. In the revised version of the manuscript we have, as you suggested, stressed the importance of supportive supervision, training, and performance review and clinical mentoring (Line # 479 and following). 

3. Reviewer comment: There is a need to better document to contribution of stockouts to lack of adherence to guidelines. Of course, better logistical support is an obvious need as well.

Response: Thank you. Details regarding facility preparedness will be reported in the forthcoming evaluation report of the OHEP intervention. 

4. Reviewer’s question: How were the 52 districts selected?

Response: As presented in the manuscript (Line # 102 and following) this was a sub-study to the baseline survey in the evaluation of the Optimizing Health Extension Program (OHEP) intervention. The 26 intervention districts were selected by the Ministry of Health in collaboration with the OHEP implementing partners. These districts were relatively low-performing as to child health and care utilization. The regional health bureaus selected 26 comparison districts, which had similar demographic and health system characteristics as the intervention districts. The researchers had no role in the selection of study districts. A full description of the intervention protocol has been published (See Line # 113). 

5. Reviewer’s question: On p. 6, lines 151-2: malnutrition, anemia, HIV and immunization/ vitamin A status are not symptoms. How anemia and HIV status are determined might be explained. Malnutrition is assessed by MUAC, of course, and immunization/vit A status were presumable assessed by asking the caretaker.

Response: Thank you, we have modified the text for clarity (Line #121 and following). 

Reviewer’s question: How was the community mobilization to get children to attend the health posts carried out?

Response: We have expanded the description of the mobilization in the revised version of the manuscript (Line # 130 and following). The mobilization was done by the data collection team in collaboration with the local administration. All families in the area with children 2-59 months old who were sick on the days of the survey were invited to come to the local health post for examination and treatment.

6. Reviewer’s question: It would be important to document if possible when the iCCM training was provided and what kind of in-service refresher training had been provided. It would also be important to know if the supervisors of the HEWs had been trained in iCCM since then? The findings suggest that this might be one practical implication of the study findings – along with more frequent in-service training on iCCM or perhaps have HEWs go through the entire training again from time to time. After all, recertification is a common requirement for many trainings in the US (cardiopulmonary resuscitation, specialty boards, etc.). What is the evidence from other studies about the “decay rate” in knowledge and practices for CHWs after iCCM? Is there any?

Response: We have added this information in the Results section (Line # 217 and following) and discussed the potential consequences of lack of recent re-training in the Discussion section (Line # 440 and following).

7. Reviewer’s question: How often were drug stockouts present?

Response: This paper mainly aimed to present how HEWs clinically assessed, classified and treated sick children. Information on the health post preparedness, including medicines, is reported in the forthcoming evaluation report of the OHEP intervention. 

8. Reviewer’s question: What are the criteria for RDT testing? Are these contextually appropriate? Should all children with fever have had an RDT for malaria?

Response: The recommendation is that children who have fever or a history of fever in the past 48 hours should have an RDT. In reality, health workers in facilities on higher altitude in the Ethiopian highlands may find this less motivated.

9. Reviewer’s question: Were Ready-to-Use Therapeutic Foods actually available at the health post?

Response: Yes, RUTF were in most cases available at the health posts and the health extension workers were trained to provide this to diagnosed malnourished children. 

10. Reviewer’s question: Don’t HEWs also treat sick children out in the home and in the community? You might at least want to mention that.

Response: Treatment of sick children is usually provided at the health post by the health extension worker. They bring some medicines and supply when visiting families and may provide treatment to sick children, but preventive activities dominate during home visits. We have added some information to the text (Line # 124-127).

11. Reviewers comment: The paper should at least mention all of the various tasks and roles that HEWs perform beyond iCCM.

Response: HEWs provides 17 packages of services and the integrated Community Case Management is one component of the clinical services under the health service extension program. We present these packages in a supplementary box, mentioned on Line # 118: S1 Box.

12. Minor suggestions for revisions

a. Reviewers comment: The “v” in vitamin A (line 122, p. 6) should not be capitalized

Response: We have corrected this in the revised manuscript.

b. Reviewers comment: The designation for HEWs is not consistent through the paper: sometimes it is “Health extension worker,” sometimes “health extensions worker,” sometimes “HEW.”

Response: We have corrected this in the revised manuscript.

c. Reviewer’s question: Did the observer also count the respiratory rate? How accurately did HEWs accurately count AND CLASSIFY the respiratory rate?

Response: The observer did not count the respiratory rate but observed what the health extension worker did and documented this. There is a separate published report on the diagnostic accuracy (reference 17, Getachew et al. 2019) that was based on re-examination of these children.

13. Additional minor comments

a. Reviewers comment: It would be interesting to know if the 9% of HEWs observed who had not received training in iCCM performed as well as those who had been.

Response: Thank you. This group would be too small to draw any conclusions based on such a stratification.

b. Reviewers comment: An interesting follow-up study would be to talk with CHWs about why they did not follow the iCCM guidelines – was it truly a lack of knowledge or other not readily apparent (to the researcher) other complicating factor?

Response: Thank you. Some qualitative studies of the health extension workers’ situation have been done and some are planned. The bottom line seems to be that much more support is needed (supportive supervision, mentoring). We discuss this in the end of the manuscript (Line 436 and following). 

c. Reviewers comment: P. 20, line 442, should be “workers’ ” not “workers”

Response: Thanks. We have corrected this.

Reviewer #2

Reviewers comment: 

• The authors have written a paper which provides an important perspective on understanding the quality of care provided by health extension workers, and considerations for improving the care quality gap. While these results could add to the body of literature on understanding and strengthening quality in hard-to-reach settings, it requires revisions and clarification of the relationship to the broader study and how this was related to an earlier publication which compared technical quality with a re-exam of the patient. The submission needs changes to improve the clarity of writing and particularly in the methods, and results and discussion detailed below. The manuscript also needs a careful review for grammar and formatting and would consider a copy-editor before resubmission; the manuscript contains multiple typos (for example, lines 62, 198), or unclear or wordy sentences (e.g., 114; 128; 135 – inconsistent capitalization/acronym usage for HEW.

Response: Thank you. We have reviewed the manuscript and addressed the grammatical errors. 

• Major and minor comments.

1. Reviewers comments: It is unclear how these findings fit into the overall "Optimizing Health Extension Program" interventions (those which you've listed do not seem directly related to this project), and the trial registration number you've given does not mention OHEP.

Response: The Optimizing the Health Extension Program (OHEP) is an intervention initiated by the Ethiopian Ministry of Health and implemented by NGO partners. It aimed to increase the utilization of sick newborn and child health services. The trial registration number refers to the evaluation of this intervention. This study was part of the baseline survey within the evaluation of the intervention. We have revised the text for clarity and added the reference of the published protocol for the evaluation (Reference 20), Line # 113.

2. Reviewer’s question: The relationship to this study and published results also needs clarification. You refer to another "recent study, based on the same larger survey as this article, [which] showed that the health extension workers’ diagnostic accuracy was low" (ref 17) which included gold standard as re-exam, yet this paper only described observation with a checklist. Why were data from the re-exam not used? For assessments, what were the differences between observation and re-exam?

Response: The published paper (ref 17) focused on the diagnostic accuracy of the health extension workers by comparing their classification of child illness with the re-examiners’. In the present manuscript we analyze the adherence to the iCCM guidelines in the assessment, classification, management and treatment of children with different presenting symptoms or complaints. We have mentioned the main result of the diagnostic accuracy paper (Line # 84-85). 

3. Reviewers question: This manuscript as "part of a larger baseline" survey, but you later say that you calculated power to "correctly assess, classify and treat diseases at baseline vs endline surveys, in intervention and comparison areas." (line 147) and use of a re-exam. Please clarify

Response: Thank you. The sample size calculation relates to the forthcoming baseline-endline evaluation of the OHEP intervention. However, we also calculated a post-hoc statistical power to this particular paper. We have modified the text for clarity (Line # 142 and following). 

4. Reviewers comment: Minor: You state that "no previous study has analyzed the entire process of managing sick children at Ethiopian health posts" (line 95), but this is only related to IMCI and children 2months-5 years. This sentence should include these caveats. on describing the symptoms most commonly presented to health posts.

Response: Thank you. The health extension workers do not follow the IMCI guidelines. They follow the iCCM guidelines for children with common illnesses 2-59 months. We have revised the statement (Line # 92-93).

Methods

5. Reviewer’s question: Methods: The selection and recruitment processes are unclear and should be consolidated. In line 127, you state "participants were invited," but earlier you state that children who showed up at the health facility were included. The paper later states that community mobilization targeted sick children to have them brought to the facility – which children were targeted and why, and how did community mobilization work? You also state that children "brought to the health post by their caregivers and the health extension workers were eligible" (137) – did HEWs select children for the study in the community and then bring them to the facility?

Response: Thank you. We have revised and consolidated the description of the recruitment of sick children, see Line # 130 and following. 

6. Reviewer’s question: In line 140, you aimed for an average of 4 sick children per health post – was that a target, or a cap, or a minimum, and what efforts were made to include 4 children? How were the children included selected (who determined they had IMCI-eligible symptoms). Were extremely sick children excluded to not delay referral if needed with the re-exam? The statement "this implies that the sample was not randomly selected to represent the regions"? (359-360). Explanation for the lower number included (620 versus 800) should be explained and if entire health posts were not included (beyond ones excluded for security). Were the final regional distribution of children (Table 1) as planned?

Response: Thank you. Based on earlier studies, we estimated that on average four children per health post could be mobilized to visit for examination. This was entirely for planning purposes. As we have explained in Line # 205-207, twenty-five clusters had shared health posts and 22 health posts were not visited by eligible children during the study period due to inadequate community mobilization. We have revised the text to address your questions. As mentioned in the “Ethical considerations”, severely ill children were referred to health centers and offered transport, if needed. As far as we know, the study activities did not cause any delays to the management of severely ill children. 

Reviewer’s question: How were HEWs selected? If there are two HEWs at the health facility, who conducted the consultation and how was that decided? Were any HEWs excluded for any reason?

Response: The health posts are staffed by two, sometimes three health extension workers. In most cases there is not more than one health extension worker at the health post a given time – the other worker is busy with community activities and home visits. The health extension worker who happened to be serving at the health post at the day of the study was observed. If two and more health extension workers were available in the health posts we included all as long as they saw a child eligible for inclusion as we explained in Line # 168-171. No health extension worker was excluded for any reason. 

7. Reviewer’s question: You report that data were collected through a structured observation checklist on tablets (160), but earlier state that "re-examiners" (146) were considered the gold standard. Did re-examination occur? If not, how did you determine whether there was a misdiagnosis? (181) The checklist should be included as well as how they determined if specific actions such as counting breaths were determined. If a child had more than one eligible complaint, were both included?

Response: Thank you. In this manuscript, we report the observed management of sick children in relation to the iCCM guidelines. These children were re-examined and the diagnostic accuracy of the health extension workers has been published (reference 17, Getachew et al., 2019). We provide the checklist/questionnaire as supplementary file. This information is provided in the revised manuscript (Line # 158-159).

8. Reviewer’s question: Who are the trained supervisors (166)? What was their role and how was it different from the data collectors?

Response: Thank you. The supervisors were health professionals with a bachelor’s degree or above. They had been trained on the data collection tools, research ethics, and field work processes. They closely monitored the overall data collection processes and provided feedback to the data collectors on a daily basis. We have described the qualifications and role of supervisors in the revised manuscript (Line # 176-183).

9. Reviewer’s question: What kind of data quality checks occurred and what were the results?

Response: Data were checked by the supervisors and by data managers after submission of data from field to the server and the central study team. Data were checked for completeness, consistency and duplication. Identified errors were corrected. We have added to the earlier description of these procedures, Line 181-183.

10. Reviewer’s question: The description of power calculation is unclear (142-147) – it states that the study was powered to "correctly assess, classify and treat diseases at baseline vs endline surveys, in intervention and comparison areas" (148) –. In line 149, you then state that the "study was powered to "ascertain prevalence (for example when performing a certain assessment)" – prevalence of what? Finally, if only 620 children ultimately included how did the power increased from 80% to 85%?

Response: Thank you. The sample size calculation relates to the forthcoming evaluation of the OHEP intervention with baseline and endline surveys. As mentioned in the text, we also did a post-hoc power analysis for this analysis and manuscript. The prevalence considered in that calculation was the assessment of clinical danger signs. We have modified this section of the text, Line # 142 and following.

11. Reviewer’s question: Ethics: It is important to note if there any considerations for delay of care for severely ill children if gold standard re-review was included (see questions above).

Response: The re-examiners were trained clinical officers. After the immediate re-examination, referrals were considered whenever required. We believe that the re-examination added value and security to the whole management of the sick children, and have no information about any delays being caused by the study procedures. On the contrary, transport was offered to severely ill children if needed (as mentioned in the Ethical considerations).

12. Reviewer’s question: How was consent gathered form non-biological parents? (187)

Response: Thank you. Children were usually accompanied by the mother or father. Sometimes another guardian, e.g., a grandparent, accompanied the sick child. Informed consent was obtained from the guardian who accompanied the child and administered by the study fieldworkers. 

13. Reviewer’s question: Were children who required urgent referral but were not referred by HEWs referred by the supervisor or data collector (312)?

Response: Thank you. Yes, as mentioned in the Ethical considerations, if the health extension worker had missed a diagnosis or need of treatment or referral this was immediately discussed and managed.

14. Reviewer’s question: Results and Discussion:-Throughout, please make it clear what is the standard of care. For example – are HEWs expected to ask about all three general danger signs (line 224), assess for chest-indrawing (225), examine for meningitis in fever cases (273) and when are they supposed to prescribe antibiotics for all cases? (237, 291)? The comparison to protocols is done better in the MUAC section (293), where you describe the expectations for HEWs as compared to their performance, although you do not mention whether correctness of MUAC procedure is assessed. While many readers may be familiar with IMCI, they may differ by country. Consider a table as a supplement

Response: Thank you. We refer to the iCCM guideline as the standard in this study. We have revised the text throughout the Result and discussion section to clarify what examination and management the Guidelines prescribe (Line # 244 and following).

15. Reviewer’s question: I was confused by the data on fast breathing. For example, in line 228-231, You report that HEWs counted fast breathing in 74% of cases and only correctly labeled fast breathing in 78% of cases with fast breathing. How was that confirmed Especially if the HEW did not perform the exam?

Response: Thank you. The health extension worker counted respiratory rate in 74% of the cases with respiratory complaints. Among those assessed with counting of respiratory rate, 33% had fast breathing. Among those labelled to have fast breathing, 78% were classified as suspected pneumonia. We have slightly edited the text to avoid misunderstanding (Line 248-252).

16. Reviewer’s question: Line 201: you note that only 91% of sick child consultations were done by HEWs – who conducted the other 9%? If the study was to evaluate HEWs were these excluded? Same question for non-IMCI conditions (ex. Burns)

Response: Thank you. What is written in the text is that 91% of the consultations were done by HEWs who were trained in iCCM. The remaining 9% of child examinations were done by HEWs who had not been trained in iCCM. The non-iCCM conditions, such as burns, were also identified in this study. But this report focused on the major iCCM-related complaints.

17. Reviewer’s question: Line 215: please clarify how were symptoms assessed? It is unclear if it is from caregiver report, examination, re-examination, or some other data collector assessment. For example – in line 215, was fever just based on complaint, evaluation by the HEW, use of thermometer, etc.?

Response: The presenting complaints or symptoms were those mentioned by the parent or caregiver when entering the health post. This was followed by assessment, classification and management by the health extension worker. The text on presenting complaints (Line 233 and following) describes this. 

18. Reviewer’s question: In limitations, I was not sure that I understand the statement line 360 you can have “no reason to believe” children represented the rural areas in the 4 regions. Can the authors add in a reference for example about the range of conditions and severity being “as expected”. Also typically limitations are towards the end of the discussions, not in the middle

Response: Thank you. We have added reference number 7 (Amouzou et al., Am J Trop Med Hyg 2016) (Line # 462) that shows a similar distribution of conditions. As suggested, we have moved a discussion on the strength and limitations of the study towards the end (Line # 450 and following) incompliance with the format recommended in most journals of health system and policy research. 

19. Reviewer’s question: Understanding variability across regions would be important and consider as well a metric of overall quality per HEWs (for those with >1 observation-were there higher and lower performers or did they all provide similar overall quality (or challenges)?

Response: Thank you. As mentioned in the text, the sample was not selected to represent the regions. The study unit in this paper was the sick child, not the health extension worker.

20. Reviewer’s question: The discussion brings in data from other studies. A brief discussion why some results differ in this study is important. In addition, please include actual numbers (line 393 described the Tanzania results as more optimal.

Response: In the Tanzanian example the health workers had been trained as part of a trial. We are pointing at the insufficient training, supervision, and mentoring of the health extension workers in our manuscript. We now include the actual number (Line # 394). 

21. Reviewer’s question: I was confused by the statement line 389 that the study was “performed when the management program was fully scaled up”-the timing of this assessment and if baseline or endline needs clarification.

Response: Thank you. As mentioned in the manuscript, the iCCM program was introduced in Ethiopia in 2010. We have reviewed the text to explicitly imply that we are referring to the iCCM program not OHEP.

22. Reviewer’s question: The statement starting line 440 was confusing and needs clarification 

Response: We have revised that section for clarity. Line # 436 and following.

23. Reviewer’s question: Tables and Figures: While I appreciate the details in the figures flow diagrams, it is hard to look across tasks and diseases and without %, hard to determine rates. It would be better to present tables of the results of HEW performance in addition to the table on counseling, as the reader gets a bit lost in the text. Table 1 could be described an in a supplement if you need more space.

Response: We have presented only numbers in the figures as the denominators change and tried to include % in the text associated with the figures. As illustrated by the Reviewers comment 15, it is easy to get confused when denominators change.

24. Reviewer’s comment: Minor comments: 25 – In your abstract, you state that low care seeking may be due to unfavorable community perceptions of quality of care provided, but you do not support with references or other with evidence in the manuscript.

Response: Thank you. We have presented the references for these statements in the Introduction Line #87-91. 

25. Reviewer’s question: 68 – Please clarify whether 37,000 HEWs have been deployed over the history of the program, or are 27,000 HEWs currently deployed?

Response: 37, 000 HEWs have been deployed over the history of health extension program

26. Reviewer’s question: 249 – How is labelling by an HEW as not visibly awake different than assessing for unconsciousness?

Response: Visibly awake means that the HEW notices that the child is awake. If the child is sleeping, the HEW tries to wake up the child, assessing that it is not lethargic or unconscious. This is normal clinical practice, not only in iCCM routine.

27. Reviewer’s comment: 233-235 – Include percent along with number of children.

Response: We refrain from doing so, as the numbers are very small.

28. Reviewer’s question: 251 –What does "examined the presence of dehydration" mean – is that through physical exam? Was the appropriateness/accuracy of the exam evaluated?

Response: Yes, it is through physical examination and the observer noted whether it was done or not.

29. Reviewer’s question: 365 – How do you know what the expected mix of complaints would be?

Response: Based on previous evidence in multiple studies and clinical experience.

---

## [Decision Letter · Decision Letter 1]

7 Sep 2020

Quality of clinical assessment and management of sick children by Health Extension Workers in four regions of Ethiopia: a cross-sectional survey

PONE-D-20-06207R1

Dear Dr. Daka,

We’re pleased to inform you that your manuscript has been judged scientifically suitable for publication and will be formally accepted for publication once it meets all outstanding technical requirements.

Kind regards,

Elizeus Rutebemberwa

Academic Editor

PLOS ONE

Additional Editor Comments (optional):

Reviewers' comments:

Reviewer's Responses to Questions

**Comments to the Author**

1. If the authors have adequately addressed your comments raised in a previous round of review and you feel that this manuscript is now acceptable for publication, you may indicate that here to bypass the “Comments to the Author” section, enter your conflict of interest statement in the “Confidential to Editor” section, and submit your "Accept" recommendation.

Reviewer #1: All comments have been addressed

2. Is the manuscript technically sound, and do the data support the conclusions?

Reviewer #1: Yes

3. Has the statistical analysis been performed appropriately and rigorously? 

Reviewer #1: Yes

4. Have the authors made all data underlying the findings in their manuscript fully available?

Reviewer #1: Yes

5. Is the manuscript presented in an intelligible fashion and written in standard English?

Reviewer #1: Yes

6. Review Comments to the Author

Reviewer #1: I have read all of the responses to the comments of the reviewers. I think the changes made in the manuscript appropriately respond to the concerns of the reviewers.

7. PLOS authors have the option to publish the peer review history of their article (what does this mean?). If published, this will include your full peer review and any attached files.

Reviewer #1: **Yes: **Henry B. Perry

---

## [Editor Report · Acceptance letter]

16 Sep 2020

PONE-D-20-06207R1 

Quality of clinical assessment and management of sick children by Health Extension Workers in four regions of Ethiopia: a cross-sectional survey 

Dear Dr. Daka:

I'm pleased to inform you that your manuscript has been deemed suitable for publication in PLOS ONE. Congratulations! Your manuscript is now with our production department. 

Kind regards, 

on behalf of

Dr. Elizeus Rutebemberwa 

Academic Editor

PLOS ONE